



# Effects of temperature and salinity on sea-spray-aerosol formation simulated with a bubble-generating chamber

Svetlana Sofieva[1, 2], Eija Asmi[1], Nina S. Atanasova[1, 2], Aino E. Heikkinen[3], Emeline Vidal[4], Jonathan Duplissy[5, 6], Martin Romantschuk[2], Rostislav Kouznetsov[1, 7], Jaakko Kukkonen[1, 8], Dennis H. Bamford[2], Antti-Pekka Hyvärinen[1], Mikhail Sofiev[1]

[1] Finnish Meteorological Institute, Helsinki, FI-00560, Finland
[2] Faculty of Biological and Environmental Sciences, University of Helsinki, FI-00790, Finland
[3] Institute for Molecular Medicine Finland, HiLIFE, University of Helsinki, FI-00014, Finland
[4] CNRS, Univ. Bordeaux, Bordeaux INP, ICMCB, F-33600, Pessac Cedex, France
[5] Institute for Atmospheric and Earth System Research (INAR), University of Helsinki, FI-00014, Finland
[6] Helsinki Institute of Physics (HIP), FI-00014, Finland
[7] Obukhov Institute for Atmospheric Physics, Moscow, Russia
[8] Centre for Atmospheric and Climate Physics Research, and Centre for Climate Change Research, University of Hertfordshire; College Lane, Hatfield, AL10 9AB, UK

*Correspondence*: Mikhail Sofiev (mikhail.sofiev@fmi.fi), Antti-Pekka Hyvärinen (antti.hyvarinen@fmi.fi) and Svetlana Sofieva (svetlana.sofieva@helsinki.fi)

**Abstract.** A new bubble-generating glass chamber design with an extensive set of aerosol production experiments is presented. Compared to the experiments described in the literature, current setup is among the medium-sized installations allowing precise control over the air discharge, water temperature and salinity. The size and material of the chamber offer variety of applications due to its portability, measurement setup adjustability and sterilization option. The experiments have been conducted in a cylindrical bubbling tank of 10 L volume filled by ~30-40 % with water of controlled salt content and temperature and covered with a hermetic lid. The chamber was used to study the characteristics of the aerosols produced by bursting bubbles under different conditions. In line with previous findings, the sea spray aerosol production was shown to depend linearly on the surface area covered by the bubbles, which in turn is a near-linear function of the air discharge through the water. Observed dependencies of the aerosol size spectra and particle fluxes on water salinity and temperature, being qualitatively comparable with the previous experiments, substantially refined the existing parameterizations. In particular, the



bubble size was practically independent from the air discharge through the water body, except for very small flows. Also, the dependence of aerosol spectrum and amount on salinity was much weaker than suggested in some previous experiments. The temperature dependence, to the contrary, was significant and consistent, with a transition in the spectrum shape at ~10 °C. Theoretical analysis based on the basic conservation laws supported the main results of the experiments but also highlighted the need of better understanding of the aerosol production from a cold water surface.

## 1. Introduction

Sea spray aerosols (SSA) emitted from ocean surface significantly affect climate but their specific role in e.g. cloud formation remain uncertain (Brooks and Thornton, 2018; Wilson et al., 2015). SSAs both scatter the incoming solar radiation and indirectly act as cloud condensation nuclei (CCN) or ice nuclei (IN) modifying cloud properties and precipitation patterns. The parameters influencing the production of SSAs include water temperature, salinity, sea state (wave direction, height and shape), wind speed and organic surface-active matter (Grythe et al., 2014; Lewis and Schwartz, 2004). Given that 70 % of the globe is covered by the oceans, the significance of SSAs is emphasized as a source of the global aerosols (Grythe et al., 2014; Lewis and Schwartz, 2004; Soares et al., 2016; Sofiev et al., 2011). Furthermore, as a consequence of climate change and the complex feedback loops, the abundance and concentrations of SSAs are expected to increase in the future (Charlson et al., 1987; Latham and Smith, 1990; Soares et al., 2016).

The main mechanism of SSAs production is bubble-mediated, when bubbles produced by breaking waves, burst on the surface (Blanchard and Woodcock, 1980). The bursting process results in two types of droplets: film and jet drops. Film drops are formed when the film of a bubble cap bursts, whereas jet drops form when a vertical water capillary collapses by the gravity. It is known that the parent bubble size determines the number of produced film and jet drops: large bubbles produce mainly film drops while small bubbles produce mostly jet drops (Woolf et al., 1987). Film drops are responsible for the major proportion (~60-80 %) of sub-micrometre particles, whereas jet drops mostly contribute to the production of supermicron particles (Cipriano and Blanchard, 1981; Wang et al., 2017). Apart from size, the two types of bubble-originated droplets also differ in their chemical composition: jet drops contain



mostly inorganic salts whereas the organic matter is mostly concentrated in film drops due to their
mechanism of formation (Burrows et al., 2014; Wang et al., 2017). Marine bacteria and viruses are,
however, found in both jet and film drops (Aller et al., 2005; Blanchard, 1989, 1978; Rastelli et al., 2017).
Another mechanism of the SSA production is the direct detachment of the water droplets from the wave
crests by wind. This mechanism produces the largest aerosols but becomes significant only at very strong
winds.

Numerous parameterizations have been proposed for describing the marine aerosol size spectra
and the flux from the sea surface (see a critical review of Lewis and Schwartz (2004) and later works e.g.
Sofiev et al. (2011)). The most-widely used approximation of marine SSA emissions was suggested by
Monahan et al. (1986), albeit the majority of modern applications combine it with later amendments
expanding the emission size spectrum towards smaller particles. A consensus regarding the sub-micron
and sub-0.1 μm aerosols production at the sea surface has been evolving along with the development of
more sensitive and accurate measurement techniques. In one of the first approximations, Rossknecht et
al. (1973) suggested an exponential shape of the marine aerosol number size distribution while only super-
micron particles were observed in the study. It has been gradually recognized, however, that particles as
small as 10-30 nm in diameter comprise the bulk of the SSA number emission, but specific shapes of the
spectrum suggested in different studies vary widely (Blot et al., 2013; Clarke et al., 2006; de Leeuw and
Cohen, 2013; Lewis and Schwartz, 2004; Mårtensson et al., 2003, 2010; Sellegri et al., 2006). The water
hydrodynamics, temperature, salinity, and wind forcing, as well as the sea water chemical composition
and surfactant concentrations can all modify the marine aerosol emission (Cochran et al., 2017;
Mårtensson et al., 2003; Sellegri et al., 2006; Tseng et al., 1992; Tyree et al., 2007). Decreasing water
temperature was suggested to shift the aerosol size distribution towards the smaller sizes (Mårtensson et
al., 2003; Sellegri et al., 2006).

When SSA is generated in laboratory conditions, the challenge is to mimic the key processes of
the real environment: bubble bursting and initial aerosol generation. Commonly applied methods include
atomizers and bubbling tanks with sintered glass diffuser devices or water jet bubbling systems
(Christiansen et al., 2019; Drenckhan and Saint-Jalmes, 2015; Fuentes et al., 2010; Leifer et al., 2003;
Mårtensson et al., 2003; Sellegri et al., 2006; Tyree et al., 2007). Aerosol atomizers, being widely used



to produce aerosol mixtures in laboratory, do not mimic the dynamics of the marine bubble bursting, whereas this process can be closer replicated in a bubbling tank (Fuentes et al., 2010). Arguably the closest reproduction of wave breaking and bubble generation processes was achieved in an ocean-atmosphere

facility of Prather et al. (2013) but the complexity and costs of the experiments were rather high.

Several designs of bubbling tanks have been presented (Christiansen et al., 2019; Fuentes et al., 2010; Leifer et al., 2003; Mårtensson et al., 2003; Prather et al., 2013; Rastelli et al., 2017; Salter et al., 2014; Schwier et al., 2015). The bubbling chamber presented by Mårtensson et al. (2003), one of most-frequently cited works in application to atmospheric modelling, was a flask with a volume of 2.0 L that

was filled with 1.0 L of water. The bubbles were generated with a sintered glass filter installed approximately 4 cm below the water surface with pore sizes of 20–40 μm. Sellegri et al. (2006) used a 30 L sealed Perspex tank that was 1/3 filled and continuously flushed with 6 L min$^{-1}$ of filtered air. They relied on two methods of bubble generation: weir created by pumping water and sintered glass filters. Tyree et al. (2007) constructed a bubbling tank that was a glass column filled with 7.2 L of water and the

bubbles generated using a fine- or a medium- pore diffuser (80 μm and 140 μm pore size, respectively). Fuentes et al. (2010) compared bubble and aerosol size distributions generated by a plunging-water jet system, porous media bubblers and an aerosol atomizer in an 11 L polytetrafluoroethylene (PTFE) bubbling tank. Christiansen et al. (2019) performed bubbling experiments in a stainless-steel cylindrical 34 L tank using two bubble generation methods, plunging jet and diffuser. The varied parameters in their

experiments were water temperature, bubble generation method, bubbling flow rate and water algal concentration. Salter et al. (2014) used 104 L stainless steel vessel coated with polytetrafluoroethylene (PTFE) below the water level and plunging jet bubble generation method to study the effect of seawater temperature on SSA production over long periods of time. The temperature was accurately controlled with a circulating water bath containing 30% glycerol. Schwier et al. (2015) studied the marine emission

of cloud condensation nuclei (CCN), distribution, and the impact of the added organics on CCN. They used a portable 10L glass tank filled with 3.6 L of seawater. The operating parameters were selected based on earlier studies (Fuentes et al., 2010; Leifer et al., 2003). Particle-free air was blown over the surface to mimic the wind effect. Many bubbling tanks included an optical system to monitor the bubble size (e.g. Leifer et al., 2003; Sellegri et al., 2006).



A large number of SSA generation experiments with a variety of different setups have not yet resulted in a consensus of the real-life processes forming the sea spray and aerosols. Conversely, the variety of the results and suggested parameterizations does not seem to be reducing with increasing number of studies, except for the demonstrated presence of the sub-0.1 µm particles. At the same time, parametrizations based on these experiments and applied to atmospheric composition models are not able to reproduce measured

variations in atmospheric SSA emission fluxes over the globe (Sofiev et al., 2011; Textor et al., 2006; Witek et al., 2016). In particular, the dependencies of the production term on water temperature and salinity have been challenged.

The aim of the current study is to re-evaluate the SSA production as a function of water parameters and verify the findings with basic analytical considerations. It also lays down the technological

background for further experiments with the organic matter and biological species injected into the air with SSA.

In this paper, we describe a new bubble-generating chamber designed to evaluate the aerosol production in marine-simulating conditions and to refine the most-important parameterizations of SSA production processes. We present a series of dedicated experiments with artificial salty water with widely

varying and tightly controlled bubbling air flow, water salinity, and temperature. To assess the effect of real water composition, two sets of experiments were conducted with water from Mediterranean and Baltic Sea. The results of the experiments are compared with theoretical expectations and other experimental works.

## 2.  Materials and Methods

The cylindrical glass bubbling tank constructed for this study has the advantages of being comparatively large but portable, autoclavable, and equipped with multiple exit and entry points for different types of measurement devices (Fig. B1).





### 2.1. Assembly of the chamber

A cylindrical glass chamber (height 320 mm, diameter 204 mm, volume 10 L) and a compatible glass lid
were custom made by Laborexin Oy. The interphase between the chamber and the lid is sealed with a
silicon gasket and a metal ring with adjustable diameter. The chamber contains two inlets on the vertical
wall (50 mm and 160 mm from the bottom) (Fig. B1). A capillary with $D_{cap} = 1.2$ mm of inner diameter
of the output nozzle is attached to the lower inlet and acts as a bubble creating nozzle. The selected
capillary parameters were optimal for single bubble formation tests and produced a narrower range of
bubble sizes compared to the sinter filters. The lid contains five inlets similar to those on the side. Particle
counters, an exhaust air collecting tube, and a flush airline are connected to the chamber via lid inlets
(Table 1). The inlets not connected to any external devices are sealed with plastic stoppers to form a
closed system.

The chamber receives purified ($2 \times \varnothing 1$ µm and $1 \times \varnothing 0.01$ µm filters, description in Appendix B) in-house
compressed dry air (7 bar), which is first decreased to 2 bar and then directed to the manifold of two
magnetic valves. The valve separates two air lines, called here bubble and flush. The bubble line is
attached to the bubble-creating capillary. The flush line attached to the lid is used to purify the system
and to maintain appropriate pressure balance. It also offers a possibility to efficiently dilute the chamber
air. Both lines contain airflow controllers, and bubble line includes an additional pressure regulator. Both
lines also include a non-return valve and a ball valve to prevent water leakage.

#### 2.1.1. Aerosol characterization with Particle counters

Four online particle counters were installed in the chamber system: a Condensation Particle Counter
(CPC), an Optical Particle Sizer (OPS), a Differential Mobility Particle Sizer (DMPS) and an Aerosol
Particle Sizer (APS) (Table 1). The OPS, DMPS and APS measure the particle number size distribution,
and the CPC measured the total aerosol concentration in the experiments. The flow rate of each instrument
was measured regularly (TSI Mass flow meter 4143), DMPS raw data was inverted to final particle size
distributions as described by Wiedensohler et al. (2012) (FMI DMPS) and compared with the total CPC
numbers. The specifications of the used aerosol measurement devices are listed in Table 1.





To ensure that only dry particles were measured, the sample air reaching the particle counters was dried

with silica-gel based diffusion driers (Topas DDU 570). The relative humidity, RH, after the driers was

monitored with Rotronic Hygroclip RH- sensors and a chilled mirror dew point sensor (Edgetech

DewMaster) to ensure RH < 30%).

The equivalent particle sizes obtained by different devices and expressed as a diameter, are not directly

comparable due to different measurement principles (see Table 1). Electrical mobility diameter measured

by DMPS is a geometric diameter assuming the spherical shape of the particles. The aerodynamic

diameter $D_a$ from the APS was converted to electrical mobility diameter $D_e$ following Khlystov et al.

(2004): $D_e = D_a \sqrt{\chi \frac{\rho_0}{\rho_p}}$

where $\chi$ is the shape factor, $\rho$ is density, and subscripts $0$ and $p$ denote to reference density and particle

density, respectively. The shape factor for sodium chloride was estimated to be 1.10 for our size range of

interest based on Wang et al. (2010), the particle density was estimated as 2.16 g cm$^{-3}$, while the reference

density was 1 g cm$^{-3}$.

To relate the optical diameter with the electrical mobility diameter, we rely on the fact that the OPS is

factory- calibrated utilizing polystyrene latex (PSL) spheres which have a refractive index of 1.588. As

this is relatively close to the refractive index of sodium chloride (1.54), and sodium chloride shape factor

is close to unity, we can assume that the difference between the PSL and the sodium chloride particle

optical diameters is negligible, but can cause insignificant discontinuity of the spectra over the

overlapping size ranges (0.3-0.6µm) (Viskari et al., 2012; Wiedensohler et al., 2012). Thus, the OPS data

were used without any further diameter conversion.



**Table 1. Particle counters and their specifications used in the experiment**

| Instrument | Measured parameter | Manufacturer, model | Size range | Sizing method | Time resolution |
|---|---|---|---|---|---|
| **CPC** | total particle concentration | Airmodus A20 | > 5 nm | - | 1 s |
| **DMPS** | number size distribution | Home made with Medium Hauke type DMA (Differential Mobility Analyzer) and TSI 3772 CPC | 10 – 600 nm | electrical mobility diameter | ~7 min |
| **APS** | number size distribution | TSI 3321 | 0.5 – 20 µm | aerodynamic diameter | 1 min |
| **OPS** | number size distribution | TSI 3330 | 0.3 – 10 µm | optical diameter (light scattering) | 10 s |

### 2.1.2. Bubble size characterization with filming cameras

Two digital cameras (Creative Live! Cam Chat HD VF0790) were installed to record both horizontal and vertical view of the bubbling in chamber. The bubble-generating air flow was sufficiently low to generate only single layer of bubbles on the surface at all tested air flows (Fig. C1). Therefore, the images could be analysed as two-dimensional still-images from vertical view camera with the third dimension always being single bubble thick. The distinctive circular shape of the interfacial bubbles allowed determination of bubble diameter by photographic methods. The still images and films were used for analysing the bubble size distributions and the foam area covering the surface (see calculations in Sect. 3.1.1, 3.1.2).

Five still images were taken per each experiment (Table 2) and the images were analysed with ImageJ software (version 1.51, National Institute of Health) (Schneider et al., 2012). The brightness and contrast were adjusted for each image to highlight the bubbles on the surface. Bubbles were characterized as circular shapes with dark outlines (Fig. C1, Appendix C). Scaling was adjusted according to the chamber diameter of 204 mm. The photographic methods for bubble size and shape determination have been compared for a range of techniques, such as the standard funnel method and the acoustic methods, generally showing good consistency (Leifer et al., 2003; Vazquez et al., 2005).





## 2.2. Experimental setup

Four different sets of experiments have been performed in the chamber (Table 2). Most of experiments were made with the chamber filled with sodium chloride (NaCl) solutions with varying operational parameters (T, salinity, and bubble flow rate). Temperature experiment included also testing the Baltic and Mediterranean sea waters. Similar protocols were applied in each of the experiments, as listed in Table 2 and described below. Prior to each experiment, the chamber was washed with tap water and detergent, then rinsed with MQ, autoclaved ultrapure water (type 1, resistivity >18.2 MΩ·cm) purified with Milli-Q® Direct 8/16 System (used with Q-PAK® TEX-, Progard® T3- and BioPak® UF cartridges), and allowed to air dry at room temperature. Even though the chamber can be autoclaved, sterilization was not required for the inorganic specimen experiments. At the beginning of each experiment, the chamber was flushed with purified in-house air for at least 30 min to remove all remaining particles from the system. After changing the controlling parameter, the system was let to equilibrate for 30 min prior to beginning the sampling. Sampling times ranged from 30 to 60 min.

The size of the chamber (Fig. B1) and the selected flow rates (Table 2) ensured that the bubble-generating air flow results in an upward particle flux in the chamber with velocity not exceeding 0.2 mm sec$^{-1}$ in our experiments. The flux speed less than 1 mm sec$^{-1}$ is lower than dry deposition velocity of any of the produced particles (Kouznetsov and Sofiev, 2012). Large diameter of the chamber allowed to neglect particle deposition on its walls.





**Table 2: Description of experiments.**

| Experiment | Description | Varying parameter | Fixed parameters | Observed parameters |
|---|---|---|---|---|
| **Bubble size experiment** | Foam on the water surface at different bubble flow rates | Flow rate: 0.01 L min⁻¹, 0.2 L min⁻¹, 0.8 L min⁻¹, 1.5 L min⁻¹ Solution: MQ; 0.1 M-0.6 M NaCl; Baltic & Mediterranean sea water | Temperature: 22 ºC | Bubble sizes on the surface, foam area |
| **Air flow experiment** | Aerosol production at different bubble flow rates | Flow rate: 0.1-1.9 L min⁻¹ | Temperature: 22 ºC Salinity: 0.2 M NaCl | Aerosol size spectrum |
| **Salinity experiment** | Aerosol production from water with different NaCl molality | Solution: MQ, 0.1 M-0.6 M NaCl | Flow rate: 0.8 L min⁻¹ Temperature: 22 ºC | Aerosol size spectrum |
| **Temperature experiment** | Aerosol production at different water temperatures | Temperature: 2-30 ºC Solution: Baltic & Mediterranean sea water, 0.1 M NaCl, 0.6 M NaCl | Flow rate: 0.8 L min⁻¹ | Aerosol size spectrum |

In the bubble size experiment, bubble production from 4 L of 0.1 M, 0.2 M, 0.3 M, 0.4 M, 0.5 M, and 0.6 M NaCl-solutions, as well as from Baltic and Mediterranean sea waters, at 0.01 Lmin⁻¹, 0.2 Lmin⁻¹, 0.8 Lmin⁻¹ and 1.5 Lmin⁻¹ bubble air flow rate was monitored with cameras. Bubble sizes were determined from still-images of the vertical camera as described in Sect. 2.1.2.

In the air flow experiment, 4 L of MQ and 0.1 M NaCl-solution were tested at bubbling air flow rates of
0.1, 0.3, 0.5, 0.7, 0.9, 1.1, 1.3, 1.5, 1.7 and 1.9 Lmin⁻¹ to examine the effect of changing air flow rate on aerosol production, comparing pure and saline water (Sect. 3.1.2.). Based on these results, a flow rate of 0.8 Lmin⁻¹ in the bubble line was selected for the following salinity- and temperature- experiments since such flow rate ensured the optimal bubble generation and aerosol release: sufficient amount of released aerosols for observing the whole particle size spectrum but still limited single-bubble-thick foam area.

In the salinity experiment, the aerosol production was measured in 4 L of MQ with NaCl concentrations varying from 0.1M to 0.6 M, at 0.1 intervals, maintaining the 0.8 Lmin⁻¹ bubble line air flow rate, at the room temperature 22°C.

The effect of varying temperature on aerosol size spectra was tested in 0.1 M and 0.6 M NaCl solutions, and Baltic and Mediterranean sea waters. NaCl concentrations were selected to be equivalent to the range
of sea water molarities, 0.1 M corresponding to the concentration of Baltic Sea (on average 10‰, Laakso



et al., 2018) and 0.6 M to that of the Mediterranean Sea (38 ‰, Borghini et al., 2014). The concentration of NaCl in the chamber was adjusted using sterile 5 M NaCl stock solution. The chamber was insulated with foam plastic in order to maintain the temperature as constant as possible while measuring. The temperature was monitored between the particle size distribution measurements. The tests were run at 2°C, 5°C, 9°C, 11°C, 13°C, 19.5°C, 25°C and 30°C. The volume of the artificial saline water was 4 L and of the natural sea waters 2.7 L. Before the experiment, 0.5 L of the water solutions were frozen to ice cubes, added to the rest of the water, and thawed immediately before sampling. The remaining 3.5 L of saline water and 2.2 L of sea waters, and the chamber, as well as the insulations were kept at 4 °C overnight prior to the experiment. As a result, the solution reached as low as 2°C, which was the first measurement point of the experiment (Table 2). Each measurement period lasted approximately 35 minutes during which the temperature rose by 1-2°C. A heating magnetic stirrer (Witeg Premium Hotplate Stirrer MSH-30D) was used for temperature control and water mixing.

Baltic Sea water was collected on 3rd of June in 2018, 8:40 UTC at Utö Island, Finland (59°46.840'N, 21°22.130'E), approximately 500 meters from the shore. The surface microlayer was collected and stored in sterile plastic bottles. Bottles were transported to Helsinki within nine hours, keeping them at room temperature, after which they were frozen. Mediterranean Sea water was collected on 7th of July in 2018 in Miami Playa, Spain (40°59'53.2"N, 0°56'04.3"E) at 13:00 UTC. At the time of sampling, the distance from the shore was 200–300 m. The bottles were frozen approximately 30 minutes after collection and transported in the frozen state to Helsinki.

## 3. Results

### 3.1. Generation of bubbles and their lifetime

Generation and lifecycle of bubbles on the surface of various liquids has been attracting attention for centuries (Maxwell, 1874; Plateau, 1873). Despite the extensive interest and developed comprehensive models, substantial uncertainty still exists, mainly owing to the extreme complexity and diversity of the governing processes (Lewis and Schwartz, 2004; Lorenceau and Rouyer, 2020; Poulain et al., 2018).



Within the current study, we concentrate only on two parameters related to the bubble production: the characteristic bubble size and the bubble foam lifetime at the water surface. These parameters are important for the follow-up construction of a physical model of the sea spray generation. Wherever possible, the experiments are presented together with basic theoretical considerations highlighting the
controlling mechanisms and suggesting the shapes of the key dependencies.

### 3.1.1.  Bubble size for different flow rates and salinities

The bubble formation and departure from the surface of the air-supply capillary, on which it is formed, are controlled by water density, surface tension, and the wettability of the surface on which the bubble is formed. If kinematic effects of the outgoing air jet can be neglected, the bubble breakout occurs when the
buoyancy lifting the bubble exceeds the surface tension force, which attaches it to the surface. Denoting the volume of the forming bubble as $V$, its diameter at the breakout plane as $D_b$, water surface tension as $\gamma$, air and water densities as $\rho_a$ and $\rho_w$, respectively, and gravity acceleration as $g$, one obtains the following relation for the breakout moment:

**(1)**
$$(\rho_w - \rho_a)gV_b = \pi D_b \gamma$$
$$V_b = \frac{\pi D_b \gamma}{(\rho_w - \rho_a)g}$$

For pure water at 293 K, $\gamma \sim 73$ mN m$^{-1}$ (Pátek et al., 2016). The shape of the detaching bubble and, consequently, the relation between $V_b$ and $D_b$ determine the final bubble volume. Here and in the below analysis, the water layer above the forming bubble is assumed thin, so that the additional pressure due to overlaying water can be neglected.

Analysis of videos of the bubble production suggested two distinct regimes of the bubble formation: (i)
slow formation of a bubble and (ii) the bubbles are formed far from the capillary due to the fast injection of air jet into the water body (the only regime with kinematic effect). The slow bubble formation regime can be altered depending on capillary configuration: bubble formation at the exit plane of the upward-





looking capillary or formation of a bubble around the exit hole of the downward/sideways looking capillary.

For the upward-looking capillary used in all experiments discussed below, the forming bubble has the horizontal diameter equal to that of the capillary whereas its volume is computed from the Eq. (1) with $D_b$ equal to the capillary diameter $D_{cap}$. The experiment was performed with $D_{cap} = 1.2$ mm, i.e., the final bubble volume and diameter (after it detaches from the capillary and gets spherical) were 28 mm$^3$ and 3.7 mm, respectively. This prediction matches closely the experimental mean size of 3.74 mm (Fig.

1, 0.01 l min$^{-1}$ flow, artificial water).

For side- or downwards-looking capillary exit, air would form a bubble near the capillary hole on the upward-looking surface of the capillary pipe. Its size would grow until the buoyancy finally takes it out. The largest bubble, the diameter of which is not limited by walls but solely controlled by the surface tension, would correspond to the bubble shape of half of the sphere attached to the capillary surface near

the air-blowing hole. The tension force then acts over the longest perimeter and holds the bubble with maximum strength. Denoting the diameter of such a half-sphere as $D$, from Eq. (1), we obtain the equation for $D$ at the breakout moment:

$$(\rho_w - \rho_a)g \frac{1}{2}\frac{\pi D_b^3}{6} = \pi D_b \gamma \tag{2}$$

Its solution for the above lab conditions leads to $D_b{\sim}9$ mm and the diameter of the spherical bubble after the detachment $D_{b\_max} = 7.5$ mm. The $D_{b\_max}$ allows substantial variation for non-horizontal surfaces: the actual shape of the about-to-detach bubble is affected by wettability and local surface inclination.



**Figure 1. Bubble sizes for different flow rates and water salinities formed from upwards-looking capillary.**

The Eq. (2) also describes the disintegration of the powerful air jet injected with high speed into the water body away from the capillari. Individual bubbles are formed dynamically but their separation from the agglomerates is presumed to be controlled by the same competition of the surface tension keeping a large air volume together and the buoyancy promoting the random fluctuations in the shape of this volume and detaching the individual bubbles from it. This semi-qualitative reasoning was confirmed in the experiment, which showed same characteristic bubble size ~7 mm for air flows of 0.2 l min$^{-1}$ and 0.8 l



min⁻¹ when the jet was sufficiently powerful but not yet producing large disturbances in the tank (Fig. 1).

Finally, the jet produced by an air flow of 1.5 l min⁻¹ (air speed in the capillary exceeding 20 m sec⁻¹)

caused sufficient turbulence to disintegrate the large bubbles broadening the size distribution towards the

small ranges and getting closer to the bubble size ranges reported for plunger systems, aquarium aerators,

and natural conditions (Deane and Stokes, 2002; Fuentes et al., 2010; Stokes et al., 2013). This effect was

visible also at 0.8 l min⁻¹ air flow, which can be considered as an upper limit of applicability of the above

models and as the optimal air flow ensuring sufficient aerosol production and acceptable bubble size

range without excessive dynamic effects.

Broadening of the distributions of Fig. 1 towards very large bubbles for powerful air flows has no relation

to the production mechanisms. It refers to coagulation of aged bubbles.

For salty water, both density and surface tension change but the variations do not exceed 10% (Kalová

and Mareš, 2018; Wang et al., 2018; Wen et al., 2018) and can be neglected, in full agreement with the

experiment, which did not show significant salinity-driven variations.

### 3.1.2. Foam area and aerosol production

The foam area obtained during the bubble generation experiment showed a nearly-linear dependence on

the bubble flow rate (Fig. 2). A more precise fitting made for all salinities (excluding the Baltic and

Mediterranean water samples) leads to a power-law relation showing some saturation:


**(3)**
$$A(F) = A_0 \left( \frac{F}{F_0} - 1 \right)^{2/3}$$

Here $F_0 = 0.01$ l min⁻¹ is flow scale and $A_0 = 0.72$ cm² is fitting constant, which together describe the

geometry of the experiment. The normalized root of mean-square error (RMSE) of the fit (3) is < 4%. For

smaller airflows approaching $F_0$, no foam is formed: the bubbles are generated slower than they break.


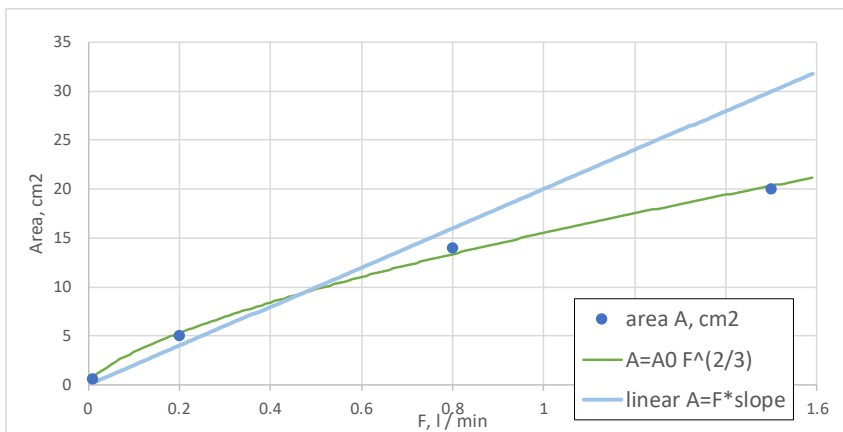

**Figure 2. Bubble foam area vs follow rate, fit of Eq. (3)**

Comparing the linear and 2/3-power fits in Fig. 2, one can see that the bubble coagulation and the foam thickening become significant only for the flow above 1 L min$^{-1}$. Omitting the $F$=1.5 L min$^{-1}$ and making a zero-intercept linear regression through the remaining points (Fig. 2) suggests the slope of 20 cm$^2$ min L$^{-1}$ = 0.033 sec m$^{-1}$. Assuming that the foam thickness $h$ is equal to the bubble diameter $d_b$ ~ 7 mm, one gets 4.7 second as a typical lifetime of the bubbles obtained in the experiment. For smaller bubbles (thinner foam, smaller foam area, lower air flow) lifetime increases, i.e., the large bubbles produced by coagulation tend to burst faster.

The above estimate, however, should be taken with caution because the dependence of the observed particle number concentration (presumably, linearly related to the foam area) on the flow rate was more complicated. In fact, the total aerosol concentration for low flow rates was practically stable whereas for high rates grew faster than linear with the flow rate (Fig. 3), with certain change of behaviour at ~ 1 l min$^{-1}$. Combined with the near-linear relation of the foam area and flow rate (Fig. 2), it suggests a reduction of the bubble lifetime with growing air flow rate. This is because for the large foam area and strong air flow, new-coming bubbles squeeze into the centre of the already existing foam, which leads to intense coagulation in the middle of the foam-covered area. In turn, coagulation results in formation of large bubbles, which, as shown above, tend to burst faster. Therefore, below, we concentrate on the experiments with the bubble flow $F$ lower but close to 1 L min$^{-1}$ and accept linear relation of the foam area and bubble air flow.





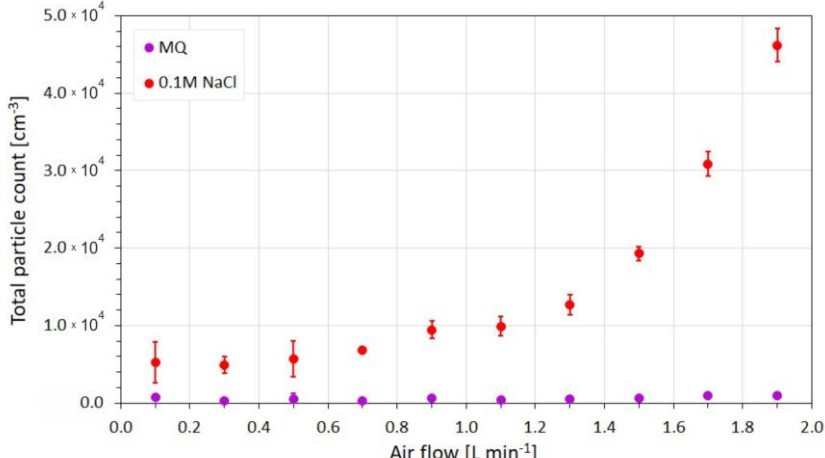

**Figure 3. Total aerosol number concentration for MQ and 0.1 M NaCl -solution as a function of the flow rate from bubble generation line.**

### 3.2. Produced aerosol size distributions

Typical size distributions obtained in the experiments have two distinct ranges: $D_p$ smaller and larger ~2 µm (Fig. 4). These ranges roughly correspond to the different mechanisms of the particle formation: film- and jet-originated bubbles (Monahan et al., 1986). The curves on Fig. 4 reveal two other peculiarities: (i) there is no reduction of the particle number concentration towards the 10 nm limit of the experimental range (ii) the MQ water, albeit showing very few particles larger than 30 nm and significantly less particles than salty water across the experimental range, still produce comparable total number of particles as the salty water. The same result was obtained for a double-distilled water (not shown) suggesting that even these artificially purified liquids still contain minuscule amounts of impurities, which do not let the droplets to evaporate completely, instead forming very small particles (Fig. 4).

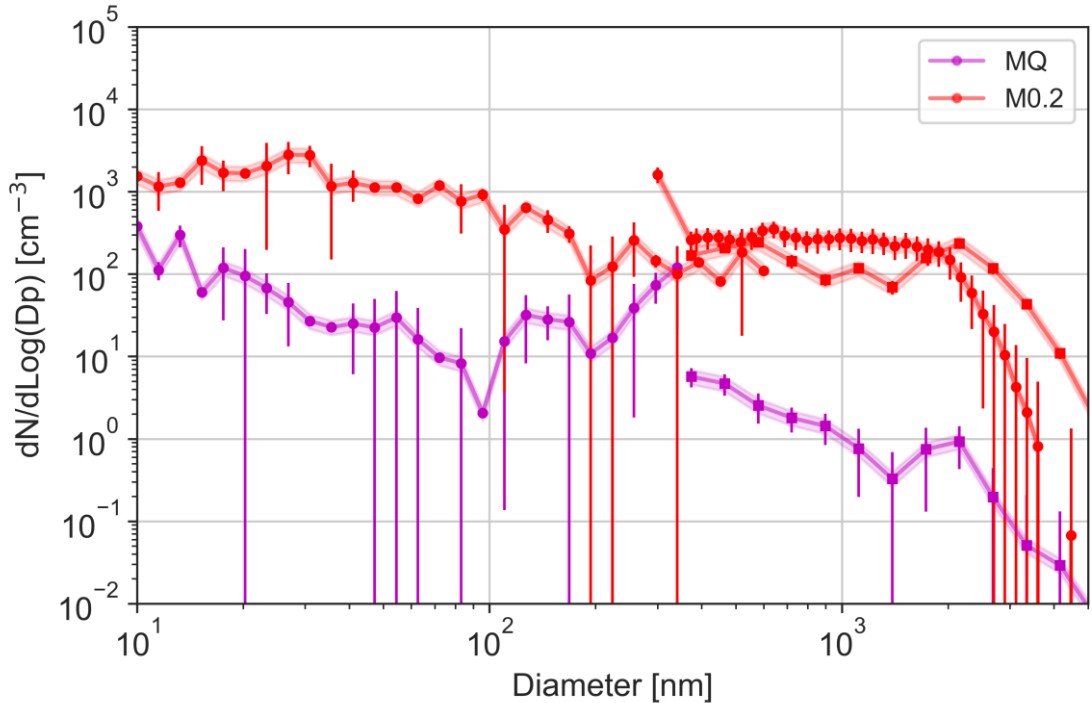

**Figure 4.** Aerosol size distribution for MQ water (lilas markers) bubbled at 0.8 l min⁻¹ and 0.2M NaCl -solution (red markers)
bubbled at 0.8 l min⁻¹; as measured with the DMPS (circles), APS (circles) and OPS (rectangles). Shading indicates pm 20% deviation
and is based on measured instability range of the system using MQ water only. Error bars show the measured standard deviation
during the experiments.

### 3.3. Impact of salinity on aerosol spectra

The observed effect of water salinity on the aerosol sizes is summarised in Fig. 5 via the bin-wise ratio of

spectra at different salinities to that of the $S=0.6M$. As one can see, the effect is quite small and not

uniform through the $D_p$ range. In particular, lower salinity leads to a clear reduction of coarse and very

small particles, whereas the intermediate-size aerosols have a tendency to grow. The amplitude of the

effect does not exceed a factor of 3-4, except for the coarsest aerosols and very low salinities. The number

concentrations of coarse aerosols ($D_p > 5$ μm) were very low (Fig. 4), so the ratios in Fig. 5 for $S < 0.3$M

should be taken with caution (the error bars are not shown for clarity of the picture).

The obtained relations are in a sharp contradiction to the dependencies suggested by a comparatively

similar but smaller-scale experiment of Mårtensson et al. (2003), who reported more than an order of

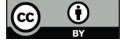

magnitude of a difference between the 3% and 0.9% salinities. However, an explanation for such a sharp effect was not provided. A simple physical mechanism of the salinity effect and its strength are presented in the Sect. 4 (Discussion).

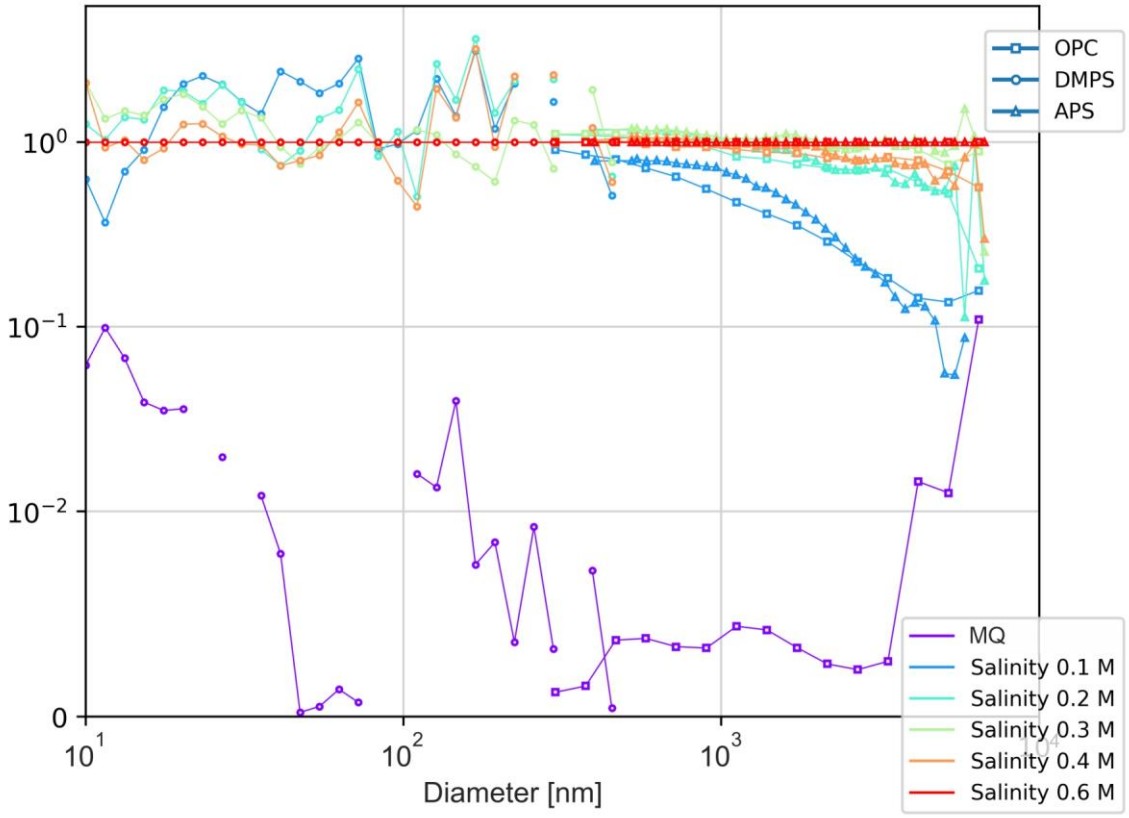


**Figure 5. Salinity effect on particle size spectrum: the spectra for different salinities normalized with the spectrum of the salinity $S=0.6\,M$**

### 3.4. Effect of water temperature on aerosol spectra

The experiments with varying water temperature (Fig. 6, Fig. 7) also brought about peculiar results observed in all experiments and confirmed in numerous repetitions. Firstly, the sensitivity of sub-100 nm particles appeared significantly higher than that of coarser ones. There was also some shift of the peak of the distribution. Secondly, the dependence exhibits a sharp change at $T_w \sim 10\ ^oC$: for warmer water the dependence is essentially negligible but for colder water the dependence is very steep: the difference in





the small particles production at 10 °C and 2.5 °C exceeds an order of magnitude. There is also a

dependence on salinity: for low-saline water the effect is smaller.

These findings are also in sharp contradiction with those of Mårtensson et al. (2003) but broadly agree with the recent results of Nielsen and Bilde (2020), who measured the total particle counts coming from burst of individual bubbles, and in excellent agreement with Zábori et al. (2012), who also noticed the 10C as a threshold of the temperature dependence. Possible reasons for the effect are discussed in the

Sect. 4 (Discussion).

Throughout almost all experiments, a substantial discontinuity was observed between the concentrations reported by DMPS and APS/OPC for the common size range of $D_p$ of 400-500 nm. The same issue is also evident in other experiments jointly using these devices (Viskari et al., 2012; Wiedensohler et al., 2012). Since the DMPS uncertainty was growing practically starting from 200 nm, we conclude that the

APS/OPC data show more accurate results. For the normalized relations (Fig. 7) such discontinuity is smaller than fluctuations of the curves themselves and therefore does not affect the conclusions of the study.

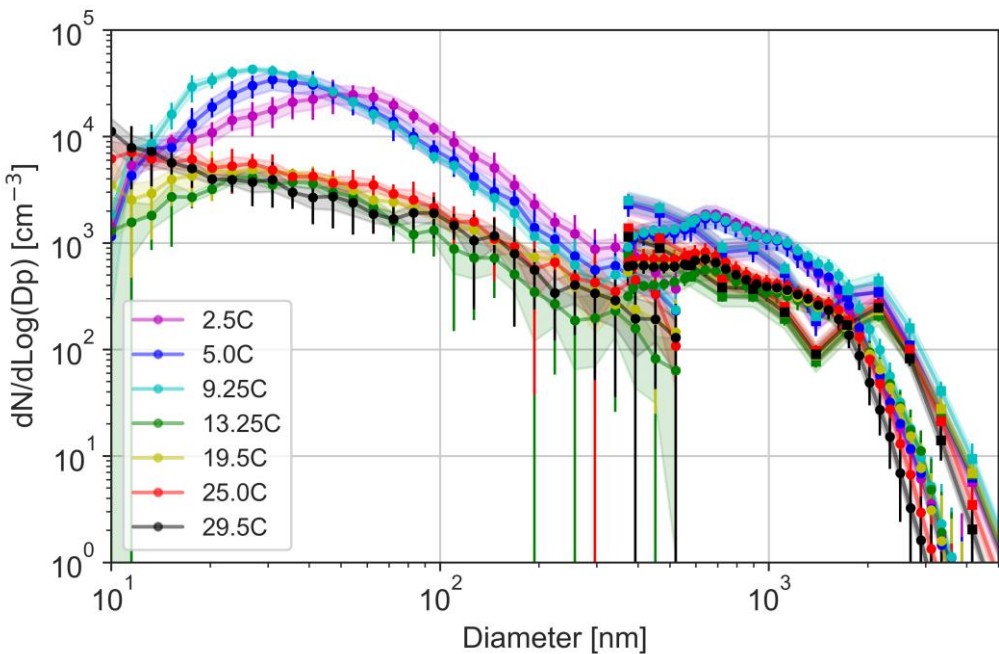

**Figure 6. Aerosol size distributions for 0.1M NaCl solution bubbled at 0.8 l min⁻¹; as measured with the DMPS (circles), OPS**

**(rectangles) and APS (triangles) for different water temperatures.**





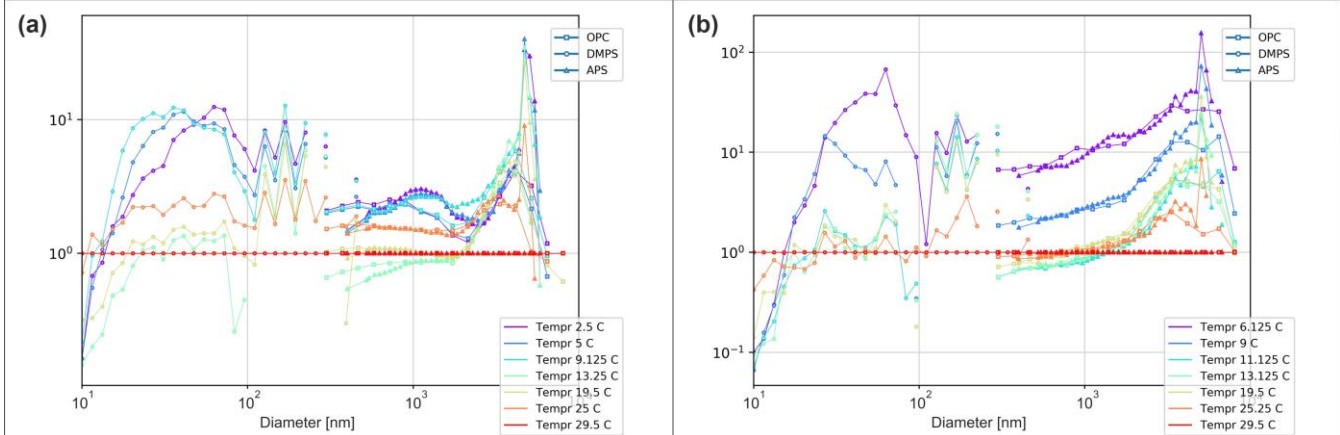

**Figure 7. Water temperature effect on particle size spectrum for salinity S=0.1M (a) and S=0.6M (b). In both panels, the spectra for different temperatures are normalized with the spectrum at T=29.5C and the corresponding salinity.**


### 3.5. Aerosol spectra for seawater samples

The experiment included two water samples collected from Mediterranean and Baltic seas. They were included in the flow- and temperature- dependencies tests (Fig. 1, Fig. 8). The sea water sessions confirmed the above-mentioned temperature dependence of the particle spectra: a sharp difference of

aerosol production for warm and cold water, with the threshold being around 10 °C. For water colder than 10 °C, production of sub-micron aerosols is much larger, and becomes sensitive to actual water temperature. However, for water warmer than 10 °C, there is practically no dependence of the particle spectrum on temperature across all sizes.






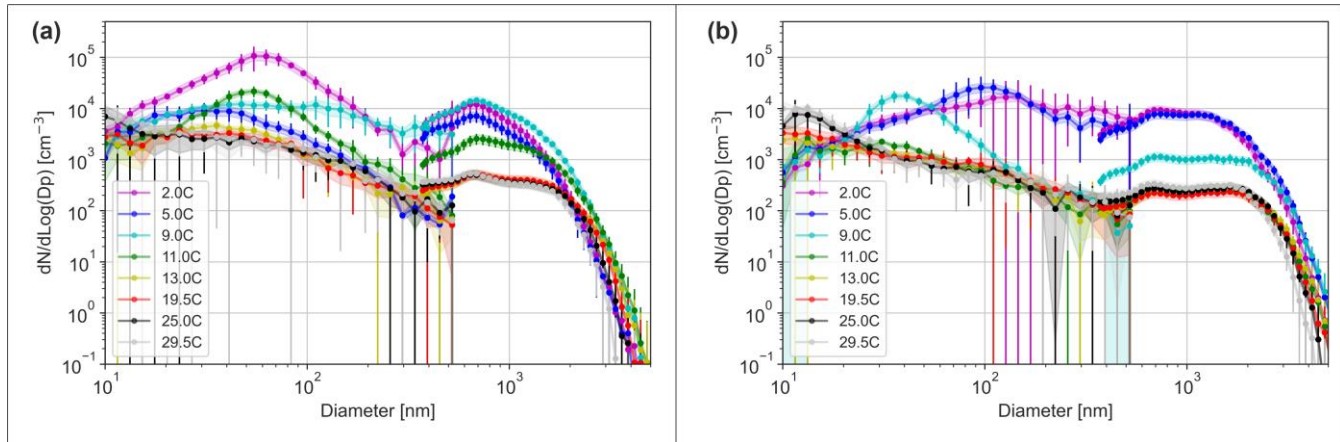

**Figure 8. Temperature- dependent aerosol size distributions for Baltic (a) and Mediterranean (b) sea water bubbled at 0.8 L min⁻¹ measured with DMPS (circles) and APS (diamonds).**

The Mediterranean Sea water particle size distribution has increased particle concentrations at sizes <1000 nm, which can be observed at low temperatures. The effect seems more pronounced than for

artificial NaCl-solutions. However, the distributions retract to a single baseline at temperatures > 11 °C. The results for the Baltic Sea water show high similarity to the results for artificial NaCl- solutions. Low temperatures exhibit increased concentrations at all size ranges, yet the peak for 100 nm<$D_p$<1000 nm is less evident.

The difference between the panels can also be related to the salinity effect: the Baltic water has about

0.93 % salt, whereas Mediterranean has ~3 %. In full agreement with the NaCl solution, the distributions and the total particle production is very similar, being sometimes even higher for the less saline water of Baltic Sea. However, direct comparison may be inaccurate because the biological content of these seas is very different, which can affect the water surface properties, bubble lifetime, and aerosol generation.

## 4.   Discussion

A usual challenge of laboratory studies is to demonstrate their representativeness for real-life conditions. It is also important to compare how the parameters, such as droplet size distribution, incidence, temperature and salinity dependency, reflect those found in natural aerosols in environmental conditions (Blot et al., 2013; Clarke et al., n.d.; de Leeuw and Cohen, 2013; Lewis and Schwartz, 2004; Mårtensson





et al., 2003; MÅrtensson et al., 2010; Sellegri et al., 2006). In the next subsections, the observed
dependencies are also verified against basic physical considerations.

### 4.1. Does the lab experiment represent the reality?

This question can be approached indirectly, by comparing with results from earlier chamber studies with
different setups (Christiansen et al., 2019; Fuentes et al., 2010; Leifer et al., 2003; Mårtensson et al., 2003;
Prather et al., 2013; Rastelli et al., 2017; Schwier et al., 2015). Supporting the representativeness of the
above results, the experiments of Fuentes et al. (2010), and Sellegri et al. (2006) confirmed that a bubbling
tank with a water jet system can closely mimic the actual oceanic distribution of the emitted bubbles and
aerosols. A similar conclusion was also made by Mårtensson et al. (2003).

An uncertainty, however, comes from the narrower range of the bubble size in this study (Fig. 1). Since
the film-droplet features are mostly determined by the bubble lifetime and produced by large bubbles
(larger than ~2mm in diameter (Lewis and Schwartz, 2004)), the bubble size should have limited effect
on the sub-μm particles. But the jet-droplets can be affected by the very low fraction of sub-mm bubbles.
That would result in lower production (a few tens of %) of droplets about 0.3-1 μm in diameter (Cipriano
and Blanchard, 1981; Deane and Stokes, 2002; Wang et al., 2017).

The effect of water composition on SSA emission is more complicated. Christiansen et al. (2019)
presented systematic variability of the produced SSA flux in relation to water temperature and bubbling
method, as well as a non-linear correlation of total particle number concentration with the water
phytoplankton mass. It is expected that the sea water chemical composition and the organic and inorganic
fractions can be significant for the bubble production method and the forming aerosols. Evidence of water
chemical composition being the controlling parameter of sea water emission was presented by Nielsen
and Bilde (2020). Therefore, confirmed replicability of our main conclusions from the artificial NaCl
solution experiments to Mediterranean and Baltic Sea water is significant.



### 4.2. The bubbles lifetime

One of the key parameters controlling the aerosol production is the lifetime of the bubbles in the foam. This is not a directly measurable parameter, but it can be derived from the bubble size and foam area. The
foam area at the water surface is controlled by the dynamic equilibrium between the bubble supply to the surface and the foam deterioration due to the bubble burst. In the experiment, the foam area was always small enough to ensure a single layer of bubbles at the surface. Then the equilibrium leads to a simple equation for the foam lifetime:

$$\frac{dA}{dt} = \frac{F}{h} - \tau^{-1}A = 0$$

(4)

$$\tau = \frac{hA}{F}$$

Here $A$ is foam area, $h$ is foam thickness, $F$ is air flow rate, $\tau$ is bubble lifetime.
Combining Eq. (3) and (4), we obtain the dependence of the foam lifetime on the air flow rate:

(5)
$$\tau = \frac{hA_0}{F}\left(\frac{F}{F_0} - 1\right)^{2/3}$$

which is applicable for $F > F_0$ when there is sufficient area of the foam to measure.

For the bubble air flow below $1\ \mathrm{l\ min^{-1}}$, a simpler linear relation (Fig. 2) leads to:

(6)
$$\tau = \frac{h\alpha F}{F} = h\alpha,$$

where $\alpha$ is the slope of the linear relation of $A$ and $F$ shown in Fig. 2.
From the Eq. (6), if the foam thickness $h$ is a constant equal to the bubble diameter in all experiments, the bubble lifetime is also the same in all experiments. For the air flows in Fig. 2, - 0.01, 0.2, 0.8 and 1.5 l min$^{-1}$ and the corresponding mean bubble size, - the lifetime will be 1.3, 0.9, 0.8, and 0.5 sec, respectively. These values, especially for the low flow rates, are very harmonious with the estimates of the detailed study of Poulain et al. (2018), who suggested a range between 0.7 and 1.5 seconds for water at the room
temperature. They also corroborate with the laboratory experiments of Anguelova and Huq (2017), who



investigated dependence of bubble properties on salinity varying over a very wide range and showed very limited dependence but substantial fluctuations of all parameters.

The Eqs.(5) - (6) can be generalized to relate the foam thickness to the flow rate. The foam gets thicker with the increase of the air flow due to bubble coagulation, the importance of which grows with the foam

area. As a result, at high flow rates, large bubbles are produced on the water surface and the bubble size distribution extends towards the large sizes, as seen in Fig. 1.

### 4.3. Bubble generation

Comparison with the bubble sizes and lifetime observed in other lab studies and in open sea shows that the results of our experiment generally agrees with other studies, but also provide some important

refinements.

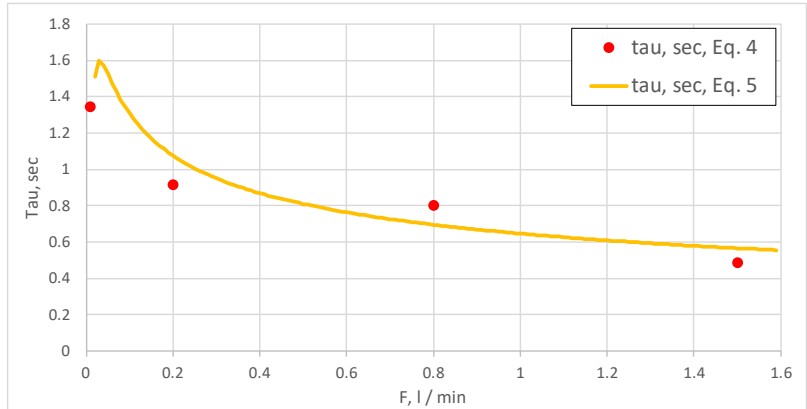

**Figure 9. Bubble lifetime, derived from the observed foam area and mean bubble size with Eq. (3), approximation of the Eq. (5) assumes the foam thickness of 7 mm.**


The data suggest that the lifetime of the bubbles is at the range of 0.5 – 1.5 seconds at room temperature (20°C), regardless the water salinity. The values for the MQ water practically do not differ from those for the MQ with added salt, coinciding with conclusions of Anguelova and Huq (2017). This result, however, is in contradiction with theoretical expectation of Lorenceau and Rouyer (2020), who argued that

individual bubbles on the surface of pure water should break within a few milliseconds. The practical





result of that study, nevertheless, showed stability of the bubble size and its dependency of the surface tension. Using an alcohol-water mixture, the authors obtained the same 7 mm for the bubble diameter for pure water ($\gamma$=73 mN m$^{-1}$) and 5.6 mm for the 12% ethanol admixture ($\gamma$=46.5 mN m$^{-1}$). The explanation for the apparent contradiction with the theoretical lifetime is probably in that the MQ water, being "clean"

from the biological standpoint, still contains substantial amount of impurities controlling the bursting process (Poulain et al., 2018).

The obtained results revealed the specificity of natural water samples, both from Mediterranean and Baltic Sea. They have much larger variability of the foam area than the artificial water. In most cases, the foam area was also larger than for artificial salty water samples. The bubble sizes showed practically no

variability being about 7 mm for practically all flow rates. Since the experiment included the artificial water samples with salinity corresponding to each of the natural samples, the large foam area and the long bubble lifetime should be attributed to the organic content in the natural water. In particular, the organics tend to form a thin film on the water surface, thus altering both surface tension and viscosity of the foam-forming water. This assumption is indirectly supported by the similarity of the bubble sizes of natural and

artificial water: the same 7 mm were reported in most experiments for all salinities. Since the bubbles are formed deep in the water, same sizes indicate that the surface tension inside the water body is not substantially different between the samples, i.e. it is the surface layer properties that control the bubble lifetime, just as observed in real-life observations.

### 4.4. Salinity effect

Water salinity can affect the particle spectra via two mechanisms. Firstly, higher salt content would result in larger crystals after the same-size droplets get dry. Secondly, the droplet sizes depend on features of the bursting bubbles, i.e., the deciding parameters will be water viscosity, surface tension, and bubble lifetime.

The first phenomenon leads to a simple relation: two identical droplets with different salt content $S_1$ and

$S_2$, would result in crystals with proportional volumes $V_1$ and $V_2$ and corresponding effective diameters $D_1$ and $D_2$:





$$\frac{S_1}{S_2} = \frac{V_1}{V_2} = \frac{D_1^3}{D_2^3}, \quad D_2 = D_1 \left(\frac{S_1}{S_2}\right)^{1/3}$$

**(7)**

Eq. (7) describes the log-homogeneous shift of the whole spectrum with regard to particle diameter. Owing to the cubic-root dependence, the effect is modest. For instance, even a change of salinity from 0.1 M to 0.8 M (already outside both the tested and realistic ranges) would just double the particle

diameters with the corresponding shift of the $\partial N/\partial log D_p$ spectrum to the right. From Fig. 4, one can see that for sub-micron particles the changes would be indeed small because the size distribution slope is small. The effect will be significant only for: (i) particles coarser than 1 μm where the slope of the distribution is large, (ii) for very low salinities, owing to the $S_2$ in denominator of Eq. (7).

The second phenomenon is more complicated but presumably small: the variations of water viscosity and

surface tension are within 10% for the realistic salinity range (Kalová and Mareš, 2018; Wang et al., 2018; Wen et al., 2018). It is also visible in Fig. 1, where bubble sizes (controlled by surface tension) show no sensitivity to salinity. The only noticeable impact could originate from bubble lifetime, which is sensitive to the abundance of the impurities in the water (Poulain et al., 2018). The higher concentration of salt should lead to the shorter bubble lifetime and the thicker bursting film of the bubbles, in turn leading to

larger and fewer particles produced by the film bursting.

These expectations agree with Fig. 5, which shows the spectra for different salinities normalized with that of $S_{ref}$= 0.6 M (the highest tested). The lower salinity indeed led to fewer coarse (> 1 μm) particles, up to a factor of a few times for S ≥ 0.2 M but with much sharper reduction for low salinities. For very small (< 20 nm) particles, there is some reduction as well, whereas the range from 20 nm to 200 nm

demonstrated some increase. In the MQ-water session, the number of particles was not enough for reliable observations, especially for coarse aerosols, which were produced in very small numbers (Fig. 4).



### 4.5. Temperature effect

The effect of temperature is arguably the most controversial in the literature. Many studies show
substantial impact of $T_w$ on aerosol production, mostly reporting a rise in colder conditions but differ
widely in details.

Our results suggested that the dependence exists only for sub-10 °C conditions with practically no effect
above that temperature. Warming the water from +2°C to +10 °C leads to 10-fold reduction of the aerosol
production. These are in close quantitative agreement with Zábori et al. (2012), who tested several
artificial solutions and Arctic Ocean water arriving at the same dependencies as registered in our
experiment: 10 °C threshold and 10-fold change of production flow between 2 °C and 10 °C. They also
reported a rise of a fraction of very small particles ($D_p$<0.25μm), in agreement with our results (Fig. 6,
Fig. 8, Fig. 10). However, the details of the size distribution spectrum were different: Zábori et al. (2012)
showed some sharp rise of the distribution at $D_p \sim$ 200 nm with about-twice lower values before and after.
They also reported a general decline of the production from 100nm towards smaller particle sizes. In our
results, on the contrary, the particles with $D_p$ of 10 nm-20 nm were dominant in almost all experiments.
Strong dependence of aerosol production on water temperature was observed by Nielsen and Bilde (2020),
who analysed individual bubbles and showed that the number of particles per bubble burst differs more
than 10-fold between 0 °C and 19 °C, varying from a factor of 2-3 up to 20-30 for different artificial
solutions and natural sea water samples. Conditions of their experiments and presentation of the results
do not allow for firm conclusions but certain non-linearity of the dependencies around 10 °C can be
noticed as well.





**Figure 10. Temperature dependence on particle concentration for aerosols sized below 100 nm (panels (a) and (d)), from 100 to 1000 nm (panels (b) and (e)), and above 1000 nm (panels (c) and (f)) for 0.1M and 0.6M NaCl –solutions ((a)-(c)) and Mediterranean- and Baltic sea water ((d)-(f)) bubbled at 0.8 l min⁻¹.**



The most-direct contradiction of our experiment and the above results are with the work of Mårtensson et al. (2003), who suggested (i) decrease of production of super-0.1 μm particles in colder water, (ii) reported the strong difference between produced aerosol size spectra throughout the whole tested temperature range: 25 °C, 15 °C, and 0 °C. The sub-zero conditions appeared the same as $T_w$=0 °C (see the analysis of Sofiev et al. (2011), who quantified these dependencies). Some elements of the experimental setup were criticized by Lewis and Schwartz (2004) and later by Witek et al. (2016), who pointed out at too high temperature sensitivity of the aerosol production, but the origin of the differences remains unexplained.

A recent experiment of Christiansen et al. (2019) compared two different ways of generating the bubbles – a diffuser, comparatively similar to our setup, and a plunger pumping the water from the bottom of the tank to its top and forcing it through a nozzle located tens of centimetres above the water level causing a waterfall. These two setups produced radically different results. The diffuser setup qualitatively agreed with our conclusions but did not show the 10 °C threshold. But plunger, showing different trends, manifested a clear minimum of production at 9-10 °C, above which the coarse particle outflow was not temperature-dependent. However, the installation used high air and water flows, thus potentially disturbing the dispersal of small aerosols and possibly adding particles produced by the waterfall itself. Bubble generation setup with plunging jet was also used in experiment of Salter et al. (2014). In their study, the ~10 ⁰C threshold was observed as a significant shift in bubble size spectra towards smaller sizes. The dependence of total particle concentration on temperature was strong below 10 ⁰C and insignificant above 10 ⁰C.

## 5. Conclusions

We have built a new sea spray production chamber and characterized its performance in a set of fully controlled laboratory experiments. Characterization was done with detailed measurements of the bubble generation and aerosol formation processes. The stability of the glass chamber was demonstrated in repeated multiple experiments varying the bubble-generating flow rate, water temperature and salinity.





The material of the chamber, its compact size and possibility to sterilize offer potential for studying sea microlayer and the effects of biological composition on SSA formation at various locations.

The flow rate-varying experiments covered the range of setups, from releasing individual bubbles one-by-one with intervals longer than their lifetime at the surface, and up to intense air flows forming air jets at the exit of the underwater capillary and wide foam at the water surface. Water and salinity experiments were performed with moderate flow rate and covered the realistic conditions: salinity from fresh (MQ) water up to 6 M of NaCl and temperature from 2 °C up to 25 °C. Experiments were also made with natural

water from Mediterranean and Baltic Seas.

The experiments quantified the dependencies of aerosol production on the main environmental conditions and manifested two important refinements, which differ from the sea salt parameterizations broadly used in the models today. In particular, they showed modest dependence of aerosol production on the water salinity (even very clean MQ water resulted in noticeable particle flow, albeit predominantly of ~10 nm

size). Secondly, the dependence on temperature manifested a saturation effect: for <10 °C cold water, lower temperature led to stronger sub-micron aerosol production, whereas above that threshold no dependence was found in any of the experiments.

The obtained dependencies were accompanied with theoretical considerations, which supported and explained the findings, also showing good quantitative agreement.

These results generally agree with recent studies on sea salt aerosol generation but point out at (i) necessity of better theoretical understanding of the differences in SSA generation in different experimental setups, which are mainly proclaimed but rarely explained; (ii) need for a review of the sea salt parameterizations currently adapted in many modern atmospheric and oceanic models.

## 6. Author contributions

E.A., N.S.A., J. D., J.K., D.H.B., A.-P. H., R.K. and M.S. planned the experiments and participated in setting up the chamber system; S.S., E.A., N.S.A., A.E.H., E.V. and A.-P. H. performed the measurements; S.S., E.A., A.E.H., A.-P. H. and M.S. analysed the data; S.S., E.A., N.S.A., A.E.H. and M.S. wrote the manuscript draft; S.S., E.A., N.S.A., J. D., M.R., J.K., D.H.B., A.-P. H., R.K. and M.S. reviewed and edited the manuscript; all authors have read and approved the paper.





## 7. Competing interests

The authors declare that they have no conflict of interest.

## 8. Acknowledgements

Thank the staff of Laborexin and Ari Salminen at SMC. Henri Servomaa, Esa Hautajoki, Virpi Mäntylä and Olli Moisio are thanked for excellent technical assistance. This study was supported by Academy of Finland Postdoctoral Grant 309570 for N.S.A. and the Academy of Finland Flagship funding (grant No. 337552) Theoretical analysis of the results was supported by the GLORIA project of Academy of Finland (grant No. 310372) and Horizon 2020 EMERGE project (grant No 874990).

## 9. Appendices

### 9.1. Appendix A. Notations

| | |
|---|---|
| $A / A_0$ | surface area covered by bubbles for a given / reference air flows $F / F_0$ |
| CCN | Cloud Condensation Nuclei |
| CPC | Condensation Particle Counter |
| $D_a$ | aerodynamic diameter of a particle |
| $D_b$ | diameter of a bubble |
| $D_{cap}$ | inner diameter of a capillary |
| $D_e$ | electrical mobility diameter of a particle |
| $D_p$ | observed dry-particle diameter after all corrections |
| DMA | Differential Mobility Analyzer |
| DMPS | Differential Mobility Particle Sizer |
| $F / F_0$ | air flow / reference air flow of the bubble generator |
| $g$ | gravity acceleration |
| $h$ | thickness of foam at the water surface |
| $\gamma$ | water surface tension |
| IN | Ice Nuclei |
| $N$ | number concentrations of particles in the air, [particles m$^{-3}$] |
| OPS | Optical Particle Sizer |
| RH | relative humidity |
| RMSE | square-root of the mean squared error |
| $\rho_a$ | air density |





| $\rho_w$ | water reference density, estimated to be 1000 kg m$^{-3}$ |
|---|---|
| $\rho_p$ | salt particle density, assumed to be 2600 kg m$^{-3}$ |
| $S$ | water salinity, NaCl concentration in the solution |
| $T$ | water temperature, °C |
| $\tau$ | foam lifetime (s) |
| $\chi$ | shape factor as defined in Khlystov et al. (2004) |
| $V_b$ | volume of bubble |

## 9.2. Appendix B. Chamber system

The chamber system presented in Fig. B1 receives inhouse compressed air produced with oil free compressor (WisAIR WIS25V, Worthington Creyssensac, Italy) and refrigeration dryer (DEiT 032, MTA S.p.A., Italy) with dew point of +3 °C (equals maximum of ~5-6 g of water per cubic meter). The compressed air passes through one ⌀ 0.01 µm filter (Friulair X Series) and two ⌀ 1 µm filters (Friulair S Series) before reaching the pressure regulator. Sequential filtering ensures the air purity, free from particles and micro-organisms. The pressure regulator lowers the pressure of incoming air from 7 bar to 2 bar, after which the air goes through HEPA-filter and reaches the manifold.

Manifold guides the air to two separate lines, flush (A) and bubble (B). Bubble line is connected to the chamber through a capillary, which is used for creating bubbling. Rate of the bubble formation can be regulated by air flow controller attached to bubble line. Flush line is connected on the chamber lid and is run for purifying the chamber and maintaining pressure balance. Air flow of the flush line is also regulated.

Total concentration and size distribution of the particles released by the bubble bursting are analyzed in optical particle sizer (OPS), condensation particle counter (CPC), Aerodynamic Particle Sizer (APS) and differential mobility particle sizer (DMPS). DMPS consists of two equipment: differential particle analyzer (DMA) and CPC. Silica driers are inserted in the system before the air reaches the particle analyzers for only calculating dry particles size and the RH meters are installed to monitor this. All the particle counters and the exhaust air tube on the lid are connected to the inhouse air removal, assuring that no air is released into the indoors.







**Fig. B1 Schematic representation of the bioaerosol chamber system.**




## 9.3. Appendix C. Determination of foam area and bubble size distribution using ImageJ 1.53K software

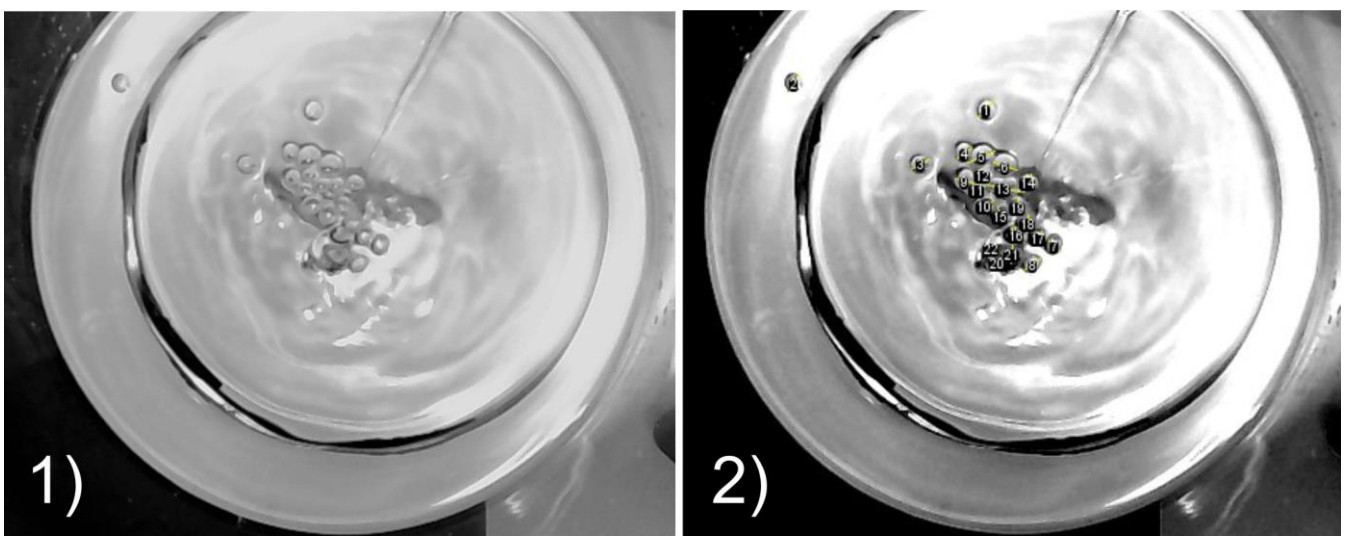

**Fig. C1. One of five images taken at 0,2 l min$^{-1}$ flow rate and $S$=0.1$M$ water salinity. 1) The initial picture from above the bubbling area. 2) Modified image with the bubble diameters selected and numbered.**

Protocol for still-image analysis:

1. Change the image type to 8-bit. Larger pixel size allows easier determination of the significant colour changes, edges of the bubbles more clearly. Adjust brightness and contrast for clarity.

2. Select the diameter of the chamber by straight line-tool, then set the scale for the diameter to be 204 mm (Analyze → Set Scale).

3. Using the ROI manager tool and the straight-line tool, select all the bubble diameters. Use "Measure"-command from the ROI tool to determine the lengths of the diameters.

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
