# Peer review of "Effects of temperature and salinity on bubble-bursting aerosol formation simulated with a bubble-generating chamber"

_Atmospheric Measurement Techniques, 2022_

## Referee Comment (RC1)

In this paper, a bubble bursting device was designed, and the effect on spray aerosol production (such as the lifetime of bubbles, bubble size, foam area, salinity, and temperature) were analyzed. Through the step-by-step adjustment of the conditions of these factors, some bubble bursting conditions are covered. And the experimental results of saltwater are compared with the seawater, which is consistent with some previous studies. This study has some serious problems that need to be addressed.

Major comments:

(1) I do not think aerosols produced from a very narrow size range of bubbles (~5-10mm) can be called sea spray aerosols. They are certainly bubble bursting aerosols, which can be related to sea spray aerosols. But sea spray aerosols are produced from a wide range of bubbles. A recent study (Jiang et al. PNAS 2022 119:e2112924119) shows submillimeter bubbles would be more important in submicron sea spray aerosol generation. I would just call the spray aerosol produced from your setup "bubble bursting aerosols". Please remove sea spray aerosol from the title and discuss how your finding can be linked to sea spray aerosol.

(2) The figures of this article do not meet the publication quality, and I hope these can be improved. (see the details in specific comments)

(2) The fluid properties of water, such as density, dynamic viscosity, and surface tension, can be changed by water temperature. Salter et al. (2014 JGRA doi: 10.1002/2013JD021376) observed that the size range of bubbles on the water surface changed significantly with temperature; for example, bubbles with a radius of less than 2 mm observed a significant decrease in number with the temperature increased. Can you have some more discussions about this impact?

I would add a figure to show the effect of temperature on the size of the bubble formation.

(3) When the temperature is above ~10 °C, the dependencies of the aerosol production on the temperature are not obvious. It seems to be inconsistent with many previous publications. Can you have some more discussions about this impact?

(4) Is the temperature of the input air stream in the chamber well controlled? Is it always at room temperature, or is it the same as the water temperature?

Specific comments:

Line 55: For the analysis of predominant (~60-80%) submicron particulates, Jiang et al. PNAS 2022 have proposed a newfound flapping mechanism that caused the film drop production in detail and can be discussed here.

Line 180: I would use the DMA selected NaCl particles to check if the OPS data can be used without any diameter conversion.

Figure 2: Please label the sub and superscript properly. Also, can you be more serious about the description of the horizontal and vertical axes?

Figure 4: Since the article has described the particle size conversion of each instrument, $D_p$ should be added to the abscissa here.

Figure 5: Why doesn't the vertical axis even have a title? Even the normalized proportions should be noted. Can the legend be put aside or some places more appropriate? I don't think the current position is proper.

Line 404: It should be "10 ℃".

Figure 6、7、8: If the color matching can be changed like this: as the temperature increases, the color of each line changes from light to dark or gradually increases to deeper color, which may be better for the trend display. Just like the salinity in Fig.5.

Line 460: In Jiang et al., bubbles smaller than about ~1 mm can predominantly contribute to submicron droplets, which can be discussed here.

Figure 9: It is recommended to write clearly about the horizontal and vertical axes. The data calculated by which equation need to correspond to the red dot and yellow line in this figure. What does it mean to write eq.4 in the red dot legend and write eq.3 in the caption?

Line 614: It should be "0.6 M of NaCl".

---

## Author Comment (AC1)

We thank the referee for the careful reading and the valuable comments, which helped to improve the paper. The referee comments are in blue, our answers are in black, and proposed modifications of text are in red.

**Major comments:**

(1) I do not think aerosols produced from a very narrow size range of bubbles (~5-10mm) can be called sea spray aerosols. They are certainly bubble bursting aerosols, which can be related to sea spray aerosols. But sea spray aerosols are produced from a wide range of bubbles. A recent study (Jiang et al. PNAS 2022 119:e2112924119) shows submillimeter bubbles would be more important in submicron sea spray aerosol generation. I would just call the spray aerosol produced from your setup "bubble bursting aerosols". Please remove sea spray aerosol from the title and discuss how your finding can be linked to sea spray aerosol.

We agree with the referee's suggestions and have modified the title and added a paragraph into discussion to take the comment into account.

The modified title is now: "Effects of temperature and salinity on bubble-bursting aerosol formation simulated with a bubble-generating chamber".

The existing discussion on relation of our results to the sea-spray formation in real atmosphere additional discussion has been extended accounting for the Jiang et al, 2022 work (Introduction and Discussion section 4.1).

(2) The figures of this article do not meet the publication quality, and I hope these can be improved. (see the details in specific comments)

The figures were updated according to the specific comments (see below).

(3) The fluid properties of water, such as density, dynamic viscosity, and surface tension, can be changed by water temperature. Salter et al. (2014 JGRA doi: 10.1002/2013JD021376) observed that the size range of bubbles on the water surface changed significantly with temperature; for example, bubbles with a radius of less than 2 mm observed a significant decrease in number with the temperature increased. Can you have some more discussions about this impact? I would add a figure to show the effect of temperature on the size of the bubble formation.

Thank you for pointing this out. Since the bubbles generated by the chamber were quite large and of welldefined size controlled by the capillary formation mechanism, we did not investigate this effect in this manuscript. It is, however, an interesting subject and we will address it in the follow-up studies, which will allow for a wider spectrum of generated bubbles.

(4) When the temperature is above  $\sim 10$  °C, the dependencies of the aerosol production on the temperature are not obvious. It seems to be inconsistent with many previous publications. Can you have some more discussions about this impact?

We agree that it is an open question of high importance. As it is already mentioned in the manuscript (Sect. 4.5), there are also numerous recent publications showing this effect of temperature. For example, in the plunging jet setup by Salter et al., 2014, the  $\sim 10$  °C threshold was observed as a significant shift in bubble size spectra towards smaller sizes, the dependence of total particle concentration on temperature was strong below 10 °C and insignificant above 10 °C. Christiansen et al., 2019, in their plunger setup, showing

different trends, manifested a clear minimum of production at 9-10 °C, above which the coarse particle outflow was not temperature-dependent. The discussion of Section 4.5 has been extended.

(5) Is the temperature of the input air stream in the chamber well controlled? Is it always at room temperature, or is it the same as the water temperature?

While testing the installation, the temperature of input air stream was controlled (the air was pre-cooled down to water temperature). The outcome was compared with room-temperature runs and no difference was found. Therefore, the main runs reported in the paper have been performed with room-temperature air,  $\sim 21^{\circ}$ C. Clarification is added.

**Specific comments:**

Line 55: For the analysis of predominant (~60-80%) submicron particulates, Jiang et al. PNAS 2022 have proposed a newfound flapping mechanism that caused the film drop production in detail and can be discussed here.

The suggested description of film drop production due to flapping mechanism was added. The existing discussion on relation of our results to the sea-spray formation in real atmosphere additional discussion has been extended accounting for the Jiang et al, 2022 work (Introduction and Discussion 4.1).

Line 180: I would use the DMA selected NaCl particles to check if the OPS data can be used without any diameter conversion.

Thank you for the advice! We shall use the approach in the future experiments.

Figure 2: Please label the sub and superscript properly. Also, can you be more serious about the description of the horizontal and vertical axes?

Labels and descriptions were added to the figure following the suggestion (see at the end of the document).

Figure 4: Since the article has described the particle size conversion of each instrument, Dp should be added to the abscissa here.

Dp was added as suggested (see at the end of the document).

Figure 5: Why doesn't the vertical axis even have a title? Even the normalized proportions should be noted. Can the legend be put aside or some places more appropriate? I don't think the current position is proper.

The legend location has been moved to a more suitable location in the figure and the axis title was added to the Y-axis (see at the end of the document).

Line 404: It should be "10 °C".

Corrected as suggested.

Figure 6, 7, 8: If the color matching can be changed like this: as the temperature increases, the color of each line changes from light to dark or gradually increases to deeper color, which may be better for the trend display. Just like the salinity in Fig.5.

Thank you for the suggestion, the colors have been updated in Figures 6 and 8. Now all the figures are consistent with each other (see at the end of the document).

Line 460: In Jiang et al., bubbles smaller than about  $\sim 1$  mm can predominantly contribute to submicron droplets, which can be discussed here.

The discussion on accounting for the Jiang et al, 2022 work has been added to both Introduction and this section

Figure 9: It is recommended to write clearly about the horizontal and vertical axes. The data calculated by which equation need to correspond to the red dot and yellow line in this figure. What does it mean to write eq.4 in the red dot legend and write eq.3 in the caption?

The legends and caption were modified to clearly describe connection between bubble lifetime calculated with equations 3 and 4 using observed foam area and bubble size, and the calculated estimation of the bubble lifetime derived from equation 5 (see at the end of the document).

Line 614: It should be "0.6 M of Na"

Corrected as suggested.

Updated tables and captions:

Table 1. Particle counters and their specifications used in the experiment. Flow rate refers to sampling flow rate of the devices.

| Instrument | Measured       | Manufacturer, model    | Size         | Sizing method      | Time       | Flow  |
|------------|----------------|------------------------|--------------|--------------------|------------|-------|
|            | parameter      |                        | range        |                    | resolution | rate  |
| CPC        | total particle | Airmodus A20           | > 5 nm       | -                  | 1 s        | 1     |
|            | concentration  |                        |              |                    |            | L/min |
| DMPS       | number size    | Home made with         | $10\ -\ 600$ | electrical         | ~7 min     | 0.7   |
|            | distribution   | Medium Hauke type      | nm           | mobility           |            | L/min |
|            |                | DMA (Differential      |              | diameter           |            |       |
|            |                | Mobility Analyzer) and |              |                    |            |       |
|            |                | TSI 3772 CPC           |              |                    |            |       |
| APS        | number size    | TSI 3321               | 0.5 - 20     | aerodynamic        | 1 min      | 1     |
|            | distribution   |                        | μm           | diameter           |            | L/min |
| OPS        | number size    | TSI 3330               | 0.3 - 10     | optical diameter   | 10 s       | 1     |
|            | distribution   |                        | μm           | (light scattering) |            | L/min |

Table 2: Description of experiments.

[revised manuscript text omitted]

---

## Author Comment (AC2)

We thank the referee for the careful reading and the valuable comments, which helped to improve the paper. The referee comments are in blue, our answers are in black, and proposed modifications of text are in red.

Overall:

Figures – I find that the quality of the figures should be improved. The figure captions should include more information – which experiments, assumptions etc. font sizes, legends should be made similar across figures.

The figures and their captions were updated to be more descriptive and uniform (see at the end of the document).

Text:

I find the manuscript somewhat difficult to follow in several places and that detailed and quantitative information is missing. I provide examples below and some suggestions for how to improve.

Improved following suggestions, see details below.

Abstract:

I recommend rewriting the abstract to provide more specific information. For example, it is not clear what is meant by "previous experiments" mentioned more than once in the abstract.

We tried to position the findings in the line of about-dozen of laboratory studies on bubble-mediated aerosol production, starting from the works of E.C. Monahan and his group in 1980s. It is difficult to mention them in the abstract due to space limitation and restricted use of references. A short clarification is added

Introduction:

Please specify what is meant by "large" and small droplets in terms of diameter ranges.

Large droplets are $>1\mu m$ and small droplets $<1\mu m$ in diameter. Clarification added to the text.

There is a long discussion in the introduction about film and jet drops – it would be relevant to relate to the current study, what kind of drops do the authors form or expect to form?

The aim was to capture both types of droplets. Clarification added at the end of Introduction where the experiment goal is announced.

"When SSA is generated in laboratory conditions, the challenge is to mimic the key processes of the real environment: bubble bursting and initial aerosol generation" I do not understand the meaning of this sentence – is bubble bursting not the initial step in bubble mediated aerosol formation?

Indeed, the repetition is unnecessary. The sentence has been corrected

Line 109: the addition of surfactants to a sea spray tank and the effect on CCN properties was also elucidated in other studies (e.g. 1-3)

The suggested studies were added to the reference list and inserted to the text.

"The effect of surfactants on CCN properties was also studied in a sea spray generation tanks by Forestieri et al. (2018), King et al. (2012) and Moore et al. (2011)."

I suggest to change the wording in line 123: "The aim of the current study is to re-evaluate the SSA production as a function of water parameters and verify the findings with basic analytical considerations." The authors present a new experimental setup and new (and interesting) data from it – these may complement existing data from other labs, but I do not really see that the authors re-evaluate something. Also I do not see that the authors "verify" the data, or at least it should be explained what is meant. As I can see, the authors interpret data. Later it says "results of experiments are compared with theoretical expectations". It is not clear to me what the theoretical expectations are.

The last two paragraphs of Introduction have been reshaped following the recommendation, shortened and unified into a single paragraph.

Line 127: what are the "most-important parameterizations" - please provide references.

The sentence has been removed leaving the specific discussions to later sections.

2.1: The lid is not shown in Figure 1.B

Lid added to the figure (see at the end of the document).

Line 135: large compared to what? It would be useful to provide some references.

Compared to the bubbling tanks presented in the literature. The sentence was modified to include reference to Introduction.

I have a few technical questions: What type of glass is the chamber made of?

Chamber is made of borosilicate glass, suitable for autoclaving. Clarification is added.

Line 142: what is the length of the capillary – or what is the distance from the tip to the water surface? What are the "selected capiliary parameters"?

The length of the capillary is 15 cm and it was always placed so that the tip of the capillary would be in the center of the chamber. The distance from tip to the water surface was 7.5 cm in experiments with MQ or NaCl-solutions and 3.5 cm in experiments with sea waters. Selected capillary parameters refers to upward-turning form and inner diameter of the capillary's output nozzle. The sentence was modified to read as follows: "…The selected capillary parameters, upward-looking nozzle and inner diameter of 1.2 mm, were optimal …".

Line 145: "compared to the sinter filters" which filters are the authors thinking about here?

The microporous borosilicate glass sinter filters such as for example DURAN® Micro Filter Candle with tube. Clarification is added.

Line 145: did the authors also test sintered filters – or is this a general comment based on literature? If so, references should be added.

Laborexin Oy, manufacturer of the glass chamber, provided us with glass sinter filters with different porosities. The tests showed the reported features. Clarification is added.

Line 153 – what is meant by appropriate pressure balance? Is it atmospheric pressure inside the chamber or is it different?

Yes, the targeted pressure inside the chamber is atmospheric pressure. The "appropriate pressure balance" now modified to "atmospheric pressure inside the chamber".

Flow rates to the instruments should be given.

Flow rates to the instruments have been added to the Table 1 (see at the end of the document).

Regarding the OPC – why did the authors not apply the shape correction of 1.1 to the data shown in the figures?

The OPC was factory-calibrated with polystyrene latex (PSL) spheres, which have similar refractive index as sodium chloride (1.588 vs 1.54) and therefore did not require shape correction or further diameter conversion.

How long time were the sea water and SML samples stored frozen before the experiments were performed? At what temperature were they kept?

The sea water samples were stored at -20 ⁰C, Baltic Sea water sample from June 2018 and Mediterranean Sea water sample from July 2018. The samples were tested in chamber system in March 2020. NaCl-solutions were prepared anew for each experiment.

Please state the supplier and purity of the NaCl used.

The supplier was Sigma-Aldrich and the purity was >99.0%.

Fig C1: I suggest the authors also show photos from the highest flow rate used to demonstrate that still only a single layer of bubbles is formed and also the area of bubbles for comparison to the 0.2 l/min case.

Photos from the highest flow rate used were added to the Figure C1.

Figure B1: please indicate the sampling flow rates of the instruments. And the ranges for the bubble generating air flow and the supplementary air added.

Sampling flow rates of the instruments and the ranges for the bubble generating- and supplementary air flow rates used in the experiments are now stated in the caption. Information is added into the table 2.

Regarding the images – five images were chosen for each experiment – why this number? - what was the time between the photos - were there variations in bubble number and foam area over the time of an experiment (30-60 min)?

Five images taken within 3 minutes from each other allowing calculation of standard deviation from the mean and were indicated in Figure 1.

It is stated that the upward particle flux in the chamber was not higher than 0.2 m/s. How was that measured /calculated?

The upward particle flux was calculated based on the size of the chamber and the input air flow to the chamber. The maximum input air flow used was 4.5 L/min and the diameter of the chamber is 204 mm, which results in upward particle flux of approximately 0.2 mm/s.

Line 215: the authors compare with a dry deposition rate "The flux speed less than 1 mm sec-1 is lower than dry deposition velocity of any of the produced particles (Kouznetsov and Sofiev, 2012)" – please explain this a bit more detailed – do the authors assume deposition onto a smooth water surface or how is this obtained?

The velocity is taken for smooth surfaces (water is smooth, even if some bubbles are on its surface), which also have the smallest deposition velocity. The sentence has been extended and clarified.

With a bubble flow rate of 0.01 L/min the dilution flow much be very high for all the instruments to get enough air? Please state the dilution flow and the flow rates to the instruments.

Yes, the dilution flow was 3.6 L min$^{-1}$. The dilution flow rates were added to Table 2.

Table 2: I suggest a column with the dilution ratio – how much air was added to the headspace in each type of experiment. This could also be included in the column Varying parameter – when the bubble flow rate is low the make up air must be correspondingly high?. I assume this dilution accounted for in all figures shown (number concentrations)?

The dilution flow rates were added to Table 2. The dilution factor was taken into account in all figures.

Line 220: it would useful also to provide the salt concentrations in g/L in all places.

Added as suggested.

Line 229: "but still limited single-bubble-thick foam area" – how large a fraction of the surface is covered in bubbles under "limited" conditions?

The relation of foam area to air flow rate from bubble line was shown in Figure 2. At selected air flow of 0.8 Lmin$^{-1}$, foam area was approximately 5% of the water surface area at any given moment.

Regarding the temperature control. It says that the temperature rose 1-2°C during each measurement. I think thus in the legends it would be more appropriate to give the temperature range for each measurement, i.e. 2.5-4.5 °C, 5-7°C, etc.

The remark has been added to the caption of Figure 7 (first figure showing the temperature experiment). Legends were kept showing mean value for the experiment to maintain the picture clarity.

Results:

It is nice that the authors provide a list of notation. I still have some questions to the notation however:

Page 12: what is "characteristic" bubble size, please define.

Bubble size characteristic for the setup used in this study.

Db is bubble diameter at the breakout plane – this should be clear also from Appendix A.

Corrected as suggested.

Dcap = 1.2 mm is the inner diameter of the output nozzle.

Modified as suggested.

How is bubble foam lifetime defined?

Foam lifetime here is the lifetime of bubbles in the foam, a parameter not measurable directly, but derived here from bubble size and foam area (see Sect. 4.2).

"These parameters are important for the follow-up construction of a physical model of the sea spray generation. Wherever possible, the experiments are presented together with basic theoretical considerations highlighting the controlling mechanisms and suggesting the shapes of the key dependencies" I do not see that the authors have constructed a general physical model of sea spray generation – or is this perhaps specifically for this setup?. I would thus suggest to rephrase this statement.

The sentence was rephrased to state the potential importance of the findings for future construction of physical models.

Line 269: The authors write: "if kinematic effects of the outgoing airjet can be neglected" – it would be nice with a bit of an explanation for this statement. Also it is later in the section discussed that kinematic effects are important at higher flow rates. Here it should be stated for which flow rates (given explicitly with values in L/min) kinematic effects are important and for which not.

In Sect 3.1.2 it was determined that the effect of kinematic effect plays a significant role at air flows above $1 \text{ L min}^{-1}$. Reference to the section and the value added to the sentence.

I suggest to show the breakup plane in the figure, and the forces acting to explain equation (1) – or alternatively provide a reference to a textbook or other where equation 1 is derived. Is the pi in equation (1) related to an angle relative to the break-out plane?

A figure added as suggested, see at the end of a document. $\pi$ in equation (1) is the mathematical constant that equals 3.1415… and is the ratio between a circle's area and the square of its radius.

The authors discuss both an upward facing capiliary (shown in fig 1B) as well as other directions – but only the upward facing is used as far as I understand and I suggest to remove discussion of the downward/sideways capilliary (e.g. line 291-302).

Removed as suggested.

I am a bit confused – in the section lines 307-315 – the authors say that they use equation 2 which is for non-upward facing capillary and refer to figure 1. At the same time they have stated that all results shown are for upward facing capillary.

Yes, the results are for the upward facing capillary, and refer to equation 1. The reference to the equation number was corrected.

Line 287: As I understand the volume obtained from equation 1 is the volume of the bubble immedialy after it has detached from the capilliary. At this point the bubble may not be spherical. The author write that the bubble gets spherical and changes it volume on the way up – how can it change its volume? The authors give a fixed value of 28 mm3 – how is that value obtained and how can it be constant across air velocity ranges?.

The volume of 28mm$^3$ is obtained from equation (1), where $D_b$, the diameter of the breakout plane, equals to the inner diameter of the capillary. The obtained value for volume does not presume that the shape of the forming bubble is spherical. However, as the bubble detaches from the capillary and moves towards the surface, it obtains spherical form, which was stated in the manuscript. Based on this assumption, the bubble diameter was calculated. Certainly, no changing volume, the sentence was corrected.

Similarly - how is the theoretical value of 3.7 mm obtained ? and how is the experimental mean size of. 3.74 mm obtained - I assume it is from the images?

The theoretical value of 3.7 is obtained from equation (1), see the clarification above. The experimental mean size was obtained from still images.

Line 290: what is "artificial water"?

NaCl solutions and MQ produced in the laboratory facility.

Figure 1: Please explain what is shown in the figure. What do the lines show (Q1 and Q3) and is it the median shown also? Are the end points min and max values? If so, why are there data points higher and lower than these?

The notations are clarified. The boxes are indeed Q1-Q3 with median shown as a horizontal dash. The upper whiskers extend from the hinge to the highest value that is within 1.5 * IQR of the hinge, where IQR is the inter-quartile range, or distance between the first and third quartiles. The lower whisker extends from the hinge to the lowest value within 1.5 * IQR of the hinge. Data beyond the end of the whiskers are outliers and plotted as points.

Line 312: what exactly is meant by characteristic bubble size ? is it from the photo's? or from calculations?

The confusing word "characteristic" has been removed.

Line 322: what do the authors mean by "aged" bubbles?

All bubbles follow three stages: formation, drainage and bursting. At the water surface, over time, the thickness of the bubble wall decreases, it drains, until the bubble cap bursts. The process of drainage is also called ageing (Poulain et al., 2018).

Figure 2: please write more information in the figure caption. What was in the tank ? water, temperature. It says that a more precise fitting was made for all salinities – why are not all the data shown in the figure

along with the precise fit? What is meant with "saturation" – does it mean the foam fills the whole surface? How do the two parameters describe the geometry of the experiment – what is their physical meaning?

The Caption was modified to include more information on experimental setup and now reads as "Bubble foam area vs flow rate, fit of Eq. (2). Mean foam area calculated from MQ and NaCl-solutions observations at +22 ºC.". The saturation refers to the decrease of curvature of the equation (2) fit. Scaling of relation bubble area and flow rate is here referred to as "geometry of the experiment".

Line 332 It would be easier for the reader if the wording was more consistent – is flow scale the same as the bubble air flow rate? Please explain the symbols. I assume A is the bubble area and F is the flow rate. But then for a flow rate of F=0 the bubble area is equal to 0.72 cm2 – and it should be zero?

$A_0$ and $F_0$ are the scaling factors. In this section, $A_0$ is the experimentally determined foam area at air flow $F_0$=0.01. Clarification added to the text.

Also – line 340 – the bubble diameter is assumed to be7 mm – but it has just been shown that it s not constant with flow rate?

Mean bubble sizes shown in Fig. 1 are constant at ~7mm in absence of kinematic effect. However the foam area, where the amount of bubbles on the surface is taken into account, changes dependant of the flow rate, which was shown in Fig. 2.

Figure 4: please explain what the "instablility range" of the system is.

It is just an uncertainty range of the chamber system. The incorrect term has been replaced.

Line 360-365: "Typical size distributions obtained in the experiments have two distinct ranges: Dp smaller and larger ~2 mm (Fig. 4). These ranges roughly correspond to the different mechanisms of the particle formation: film and jet-originated bubbles (Monahan et al., 1986)." There is more recent work4 showing that indeed jet drops do also produce a significant amount of sub-micron sized droplets. This would perhaps be interesting for the discussion.

A short discussion is added.

It should be stated how the authors see the two size regimes – I assume it is the decreasing number with size above 2 micrometer? Please explain/discuss. Why would such a trend relate to film and jet drops?

Indeed, the split was just by the slope of the curve. But we do not claim that this trend is related to the film/jet mechanisms, we only point out that the breakpoint coincides with the border between these mechanisms reported in literature.

Regarding production of particles from MQ water – it is known that atomizing MQ or otherwise purified water forms particles when atomized. LaFranchi et al. provided one possible explanation5. Perhaps it is relevant to note.

Thank you for the reference! Included.

Regarding the data on salinity it would be relevant to compare with the recent work of Zinke et al6.

This reference is, so far, a grey literature – the work has been submitted to JGR but not (yet?) accepted. Our results differ from the previous work of that group (see the Discussion section on the salinity impact), we now added comparison with other works published works.

Figure 6: perhaps it is the quality of the figure – but I do not see any triangles (APS) in the figure?

The APS data is also in the figure, but in circles. Caption corrected.

I suggest the authors expand a little on what can be concluded from Figure 6. For a given salinity the formation of particles seem to increase with decreasing temperature and more so for the smallest and largest particles? This could be discussed in relation to observations already reported in the literature. It is broadly the same trend for the two salinities?

The descriptions of the figures has been extended.

Section 4.1: I find this section somewhat speculative in several places with quite bold statements: "Supporting the representativeness of the above results, the experiments of Fuentes et al. (2010), and Sellegri et al. (2006) confirmed that a bubbling tank with a water jet system can closely mimic the actual oceanic distribution of the emitted bubbles and aerosols. " and "Therefore, confirmed replicability of our main conclusions from the artificial NaCl solution experiments to Mediterranean and Baltic Sea water is significant." Regarding the discussion on bubble lifetimes – the lifetime may depend on many factors and be very different in pure water and salt water. It would be interesting if the authors could comment on that.

The section has been reviewed towards more moderate statements about the representativeness of the results.

Figure 9: what are the uncertainties in the bubble lifetimes? I suggest to show these on the plot.

Inserted as suggested.

Line 533-536: I think the text should be rephrased for clarity – two droplets are not identical if they have different salt content? Also do the authors mean "dry" diameter rather than "effective"? Would density of the sea water also be a parameter to consider?

Corrected as suggested.

In general, when comparing with literature it should be made clear which studies have used artificial seasalt and which NaCl. Also the bubble generation method: single bubble studies, bubbles generated with a frit/diffuser and with a jet may give very different results due to the generation method. This work represents bursting of a single layer of coagulated bubbles as I understand. It is touched upon in 4.5 but in general some thoughts on how bubble lifetimes, size and particle production are expected to be comparable or different with the method used in this study from other methods would be relevant and interesting.

References to the details of the quoted studies have been added to relevant parts of Discussion. However, in the analysis we tried to refer to the features registered at more than one setup. In such cases, details of particular experiments become not so important and can be skipped.

Christiansen, S., Salter, M. E., Gorokhova, E., Nguyen, Q. T., & Bilde, M. (2019). Sea Spray Aerosol Formation: Laboratory Results on the Role of Air Entrainment, Water Temperature, and Phytoplankton Biomass. *Environmental Science & Technology*, *53*(22), 13107–13116. https://doi.org/10.1021/acs.est.9b04078

Poulain, S., Villermaux, E., & Bourouiba, L. (2018). Ageing and burst of surface bubbles. *Journal of Fluid Mechanics*, *851*, 636–671. https://doi.org/10.1017/jfm.2018.471

[revised manuscript text omitted]